# Carotenoid Cleavage Dioxygenase Gene *CCD4* Enhances Tanshinone Accumulation and Drought Resistance in *Salvia miltiorrhiza*

**DOI:** 10.3390/ijms252313223

**Published:** 2024-12-09

**Authors:** Qian Tian, Wei Han, Shuai Zhou, Liu Yang, Donghao Wang, Wen Zhou, Zhezhi Wang

**Affiliations:** Key Laboratory of the Ministry of Education for Medicinal Resources and Natural Pharmaceutical Chemistry, National Engineering Laboratory for Resource Development of Endangered Crude Drugs in Northwest of China, Shaanxi Normal University, Xi’an 710062, China; tianqian123@snnu.edu.cn (Q.T.); whan@snnu.edu.cn (W.H.); szhou@snnu.edu.cn (S.Z.); l-yang@snnu.edu.cn (L.Y.); wangdonghao@snnu.edu.cn (D.W.)

**Keywords:** *SmCCD4*, tanshinones, phenolic acid, abscisic acid, drought tolerance

## Abstract

Danshen (*Salvia miltiorrhiza* Bunge) is a perennial herbaceous plant of the Salvia genus in the family Lamiaceae. Its dry root is one of the important traditional Chinese herbal medicines with a long officinal history. The yield and quality of *S. miltiorrhiza* are influenced by various factors, among which drought is one of the most significant types of abiotic stress. Based on the transcriptome database of *S. miltiorrhiza*, our research group discovered a carotenoid cleavage dioxygenase gene, *SmCCD4*, belonging to the carotenoid cleavage oxygenase (CCO) gene family which is highly responsive to drought stress on the basis of our preceding work. Here, we identified 26 CCO genes according to the whole-genome database of *S. miltiorrhiza*. The expression pattern of *SmCCD4* showed that this gene is strongly overexpressed in the aboveground tissue of *S. miltiorrhiza*. And by constructing *SmCCD4* overexpression strains, it was shown that the overexpression of *SmCCD4* not only promotes the synthesis of abscisic acid and increases plant antioxidant activity but also regulates the synthesis of the secondary metabolites tanshinone and phenolic acids in *S. miltiorrhiza*. In summary, this study is the first in-depth and systematic identification and investigation of the CCO gene family in *S. miltiorrhiza*. The results provide useful information for further systematic research on the function of CCO genes and provide a theoretical basis for improving the yield and quality of *S. miltiorrhiza*.

## 1. Introduction

*S*. *miltiorrhiza* is a famous member of the Lamiaceae which has the effect of treating cardiovascular and cerebrovascular diseases such as coronary heart disease and myocardial infarction [1,2]. On 9 December 1997, Compound Danshen Dropping Pills passed the clinical drug application of the US FDA for the first time as a drug, achieving a historic breakthrough in the official entry of traditional Chinese medicine into the international pharmaceutical mainstream market, and was listed as one of the essential drugs for clinical emergency treatment by the country. Fang et al. [3] found that tanshinone Ⅱ A can inhibit the expression of the biomarker of cardiovascular disease (PYX3) by inhibiting the NF-κB and p38 MAPK signaling pathway to play the role of anti-atherosclerotic (AS). After giving a 0.78 mL/kg Danshen injection to streptozotocin-induced diabetes rats for 8 weeks, it was observed that the injection improved 24 h urinary protein excretion, serum creatinine, blood urea nitrogen, and other physiological functions in rats and alleviated the ultrastructural abnormalities of glomerular hypertrophy and fibrosis [4]. Danshen injection combined with clopidogrel can improve clinical efficacy in the treatment of coronary heart disease; regulate the levels of NO, thromboxane B2 (TXB2), and endothelin-1 (ET-1); promote endothelial function recovery; and have better therapeutic effects than using clopidogrel alone [5,6]. Due to its significant medicinal value, high conversion efficiency, and published genomic data, it is becoming a model plant system for studying the regulation of secondary metabolites. As shown in Table 1, the medicinal active ingredients of *S. miltiorrhiza* can be mainly divided into two categories: one includes lipophilic tanshinone compounds, including tanshinone I, tanshinone IIA, tanshinone IIB, and cryptotanshinone; the other includes water-soluble phenolic acid compounds, rosmarinic acid, salvianolic acid A, and salvianolic acid B [7]. These two types of compounds mainly accumulate in the roots of *S. miltiorrhiza* [8,9]. At present, with the extensive development of wild resources, the growth environment of *S. miltiorrhiza* has been severely damaged. Planting *S. miltiorrhiza* in arid areas is one of the important means to alleviate its resource shortage problem.

Carotenoid cleavage oxygenase (CCO) can specifically catalyze the cleavage of conjugated double bonds in carotenoid polyene chains, thereby forming various deprotonated carotenoids and their derivatives [10,11,12]. According to whether the substrate forms an epoxide structure, CCO can be further divided into carotenoid cleavage dioxygenase (CCD) and 9-cis-epoxycarotenoid dioxygenase (NCED) [13]. CCD can specifically cleave the double bonds of carotenoids, producing various aromatic compounds and pigments, as well as some plant hormones unicyclic lactones strigolactones (SLs) [14,15]. The cleavage of 9-cis-epoxycarotenoids to xanthoxin catalyzed by NCED is considered to be the key regulatory step of abscisic acid (ABA) biosynthesis [16,17]. CCO is a very ancient gene family. In 1997, the first CCO gene related to abscisic acid biosynthesis, *VP14*, was identified in the seeds of viviparous maize (*Zea mays*) with ABA-deficiency mutations [18]. Subsequently, nine homologous genes of *VP14* were identified in *Arabidopsis thaliana* [19]. In recent years, Wang et al. has further divided eight soybean CCO genes into five subgroups, namely, CCD1, CCD4, CCD7, CCD8, and NCED [20]. However, Wei et al. further divided CCO genes from different species into six subgroups, namely, CCD1, CCD4, CCD7, CCD8, NCED, and CCD-like in 2016 [21].

*CCD1* can cleave carotenoids in flowers and fruits to form aromatic compounds [22,23]. For example, LeCCD1 symmetrically cleaves multiple carotenoid substrates at the 9,10 (9′,10′) positions to produce a C14 dialdehyde and two C13 cyclohexones that vary depending on the substrate in *Lycopersicon esculentum* [24]. *CCD4* regulates the homeostasis of carotenoids, and its expression can lead to the breakdown of carotenoids, thereby regulating the color of plants’ flowers, fruits, etc. [25,26]. Some members of the CCD family can also regulate plant growth and development, such as inhibiting AcCCD8 expression in kiwifruit, which can affect plant branch development and delay leaf senescence [27]. The ntccd8 mutants in tobacco (*Nicotiana tabacum* L.) had increased shoot branching, reduced plant height, an increased number of leaves and nodes, and reduced total plant biomass compared with wild-type plants [28]. *CCD7* and *CCD8* have also been shown to be essential genes for the biosynthesis of SLs [29]. Research has shown that members of the CCO family are widely involved in various stress responses of plants. For example, in *Brassica oleracea*, *CCD1* (Bol044673) and *CCD4* (Bol009345 and Bol029878) are highly responsive to drought and salt stress [30]. The expression of the saffron *CCD4* gene can also be highly induced under drought conditions [31]. From this, it can be seen that CCDs not only affect plant odor and color and participate in the synthesis of hormones such as ABA and SLs but also regulate plant responses to various stress conditions. Abscisic acid (ABA) plays a very essential role in enhancing plant stress resistance. When plants are subjected to biological stress such as low temperature and drought, the content of ABA in the plant notably increases, causing changes in physiological and biochemical responses, thereby enhancing the plant’s stress resistance [32]. Research has shown that *A. thaliana* can highly induce the expression of related CCO under drought stress, and its overexpression leads to the accumulation of ABA [33]. In addition, the upregulation of *LeNCED1* expression in *Lycopersicum esculentum* leaves also increases the seed dormancy rate [34]. These studies suggest that CCO genes can affect the ABA-mediated plant stress response by regulating ABA synthesis.

The carotenoid cleavage dioxygenase *SmCCD4* is highly responsive to drought stress on the basis of our preceding work based on the transcriptome database of *S. miltiorrhiza*. Furthermore, all members of the CCO family in *S. miltiorrhiza* were screened and structurally analyzed. Finally, a total of 26 SmCCO genes were characterized according to the whole-genome database of *S. miltiorrhiza*. The expression pattern of *SmCCD4* indicated that this gene is strongly overexpressed in the aboveground tissues of *S. miltiorrhiza* and showed obvious responses to ABA treatment and drought stress. The overexpression of *SmCCD4* not only promotes the synthesis of ABA and increases plant antioxidant activity but also regulates the synthesis of the secondary metabolites tanshinone and phenolic acids in *S. miltiorrhiza*.

## 2. Results

### 2.1. Identification and Sequence Analysis

A total of 26 CCO family members were identified in the *S. miltiorrhiza* genome database, with 1500 bp upstream of the start codon as the promoter sequence. The length of the gene, the coding sequence and protein, the isoelectric point (PI), and the molecular weight of the SmCCO gene family members are shown in Appendix A. The length of the SmCCO gene ranges from 252 bp to 6508 bp. The length of the SmCCO CDS and protein ranges from 252 bp and 83 aa to 2409 bp and 82 aa. Among them, seven genes have the same length as the CDS, indicating that these genes only have exons and no introns. The predicted MW of the proteins ranges from 9.61 kDa to 88.49 kDa, and the PI ranges from 4.55 to 9.45. As shown in Figure 1A, a neighbor-joining phylogenetic tree of 26 SmCCO protein sequences using default parameters was constructed. According to the phylogenetic analysis, 26 SmCCO proteins could be classified into six groups. The gene structure analysis of SmCCO is shown in Figure 1B. There are a total of seven genes with only one exon and no introns, while the number of introns in other genes ranges from one to three. The result of the motif analysis is shown in Figure 1C. SmCCO members in the same group have similar motif composition.

### 2.2. Cis-Acting Elements Analysis

The cis-acting elements in the 1500 bp upstream promoter region of all SmCCO genes were analyzed by PlantCARE, as shown in Figure 2. Most elements are associated with hormonal and abiotic/biological responses. In plants, there are two types of cis-acting elements involved in transcriptional regulation in response to drought stress: ABRE and DRE [35]. Among them, ABRE is a homeotropic element that responds to abscisic acid and is possessed by ABA-dependent genes [36]. DRE is an ABA-independent gene that responds to dehydration and homeopathy [37]. According to the analysis, 17 out of 26 genes in the SmCCO gene family contain ABRE, and 2 genes contain DRE. Among them, SmCCD4 is an ABA-dependent gene and acts on drought stress.

### 2.3. Characterization and Expression Profile of SmCCD4

Many studies have shown that *CCD4* regulates the homeostasis of carotenoids, and its expression can lead to the breakdown of carotenoids, thereby regulating the color of plants such as flowers and fruits. And it is highly responsive to abiotic stress in plants. The phylogenetic analysis of the *SmCCD4* gene and the *A. thaliana* CCO gene family revealed that *SmCCD4* and *AtCCD4* are clustered together as orthologous genes. And the *SmCCD4* gene has a structural domain, RPE65 (retinal pigment epithelial membrane protein), unique to the CCO gene family, located at 100–450 bp of the gene. The coding region of the *SmCCD4* gene is 1101 bp, encoding 367 amino acids, with an isoelectric point of 5.36 and a molecular weight of 40.03 kD. As shown in Figure 3, the expression levels of *SmCCD4* gene in different tissues and under different treatment conditions were analyzed by RT-qPCR. Compared with the expression level of *SmCCD4* in roots, the expression level in the aboveground part of *S. miltiorrhiza* is significantly higher than that in roots, especially in flowers. It is obvious that hormone treatment and drought stress can induce *SmCCD4* to produce different levels of response. Under ABA induction, the expression level of the *SmCCD4* gene was the highest at 0.5 h, which was 2.7 times higher than that at the control time of 0 h. Under the stress treatment of PEG8000 solution, the expression level of *SmCCD4* increased sharply at 0.25 h, which was 2.7 times higher than that at 0 h.

### 2.4. Acquisition and Validation of SmCCD4 Transgenic Strain

We constructed the *SmCCD4* overexpression vector by using Gateway technology and then obtained *SmCCD4* transgenic overexpression plants of *S. miltiorrhiza* (OE) through A. tumefaciens-mediated leaf disc transformation. The 926 bp fragment of the *CaMV35S* promoter was detected by PCR in different strains (Appendix A). The relative expression levels of *SmCCD4* in the WT and eight OE strains are shown in Appendix A. The expression levels of *SmCCD4* in the OE strains was significantly higher than that in the WT, especially in OE-2, OE-5, and OE-8. However, considering the need for sufficient plant materials for the detection of secondary metabolite content and physiological indicators, we ultimately chose three strains, OE-4, O-6, and OE-7, for subsequent experiments.

### 2.5. SmCCD4 Regulates Tanshinone and Phenolic Acid Biosynthesis in S. miltiorrhiza

To investigate the role of *SmCCD4* in plant growth, we overexpressed *SmCCD4* in the *S. miltiorrhiza* WT strain under the control of the *CaMV35S* promoter. Due to the significantly higher expression level of *SmCCD4* in the aboveground part of *S. miltiorrhiza* compared with the roots, subsequent experimental materials were selected from the aboveground part of plants. The concentrations of four types of tanshinones in the OE strains were significantly higher than those in the WT (Figure 4A). Among them, tanshinone Ⅰ increased by 3.5 times in OE-6, tanshinone IIA increased by 6.0 times in OE-4, dihydrotanshinone increased by 2.5 times in OE-6, and cryptotanshinone increased by 5.9 times in OE-6. Compared with the WT, the content of rosmarinic acid increased by 1.4 and 1.5 times in OE-4 and OE-7 respectively, while the content of salvianolic acid in the OE strains showed no significant difference (Figure 4B). The transcription levels of seven genes, *SmDXS1*, *SmHMGR1*, *SmGPPS*, *SmGGPPS*, *SmCPS1*, *SmKSL1*, and *SmCYP76AH1*, selected from the tanshinone biosynthesis pathway were evaluated by using RT-qPCR (Figure 5A). Except for *SmGPPS1* and *SmKSL1*, which showed no significant difference in expression levels among different strains, the other genes were highly expressed in the OE strains, especially *SmCPS1*. On the contrary, the expression level of *SmCYP76AH1* in the OE strains was suppressed. Seven genes, *SmPAL1*, *SmC4H1*, *Sm4CL1*, *SmTAT1*, *SmHPPR1*, *SmRAS1*, and *SmCYP98A14*, were selected from the phenolic acid biosynthesis pathway (Figure 5B). The expression levels of *SmPAL1* in the phenylpropanoid metabolic pathway, *SmHPPR* in the tyrosine metabolic pathway, and *SmRAS* in two key downstream pathways were increased compared with the WT. The two genes *Sm4CL1* and *SmTAT1* were strongly suppressed in the OE strains.

### 2.6. SmCCD4 Overexpression Increases ABA Accumulation and Improves Drought Tolerance in S. miltiorrhiza

The content of ABA was detected in *SmCCD4* OE strains and WT strains of the control group. As shown in Figure 6A, the content of ABA in the OE strains significantly increased, especially in OE-6 and OE-7. In order to further investigate the drought resistance function of *SmCCD4*, plant materials grown for 30 days after drought treatment with 2.5% concentration of PEG8000 were collected. The contents of MDA, H_2_O_2_, CAT, and SOD in the leaves were measured (Figure 6B). Under drought stress, MDA and H_2_O_2_ contents in transgenic plants were obviously lower than those in the wild type, while SOD activity was higher than that in the wild type, and CAT activity showed no significant difference among different strains. The aforementioned results reveal that the overexpression of *SmCCD4* enhances drought tolerance in *S. miltiorrhiza*.

## 3. Discussion

With the deepening of research on the pharmacological effects of *S. miltiorrhiza* [2,38,39,40], its commercial value is being gradually recognized, and the market demand is constantly increasing. However, wild resources and existing artificial cultivation yield and quality are unable to meet the demand [41]. Expanding the planting area of *S. miltiorrhiza* and increasing the content of medicinal active ingredients are very necessary. Drought stress is a common type of stress faced by plants [42], and planting *S. miltiorrhiza* in arid areas is one of the important means to alleviate its resource shortage problem. As an important enzyme gene for carotenoid catalysis and ABA synthesis, research has shown that the CCO gene family plays an important role in regulating plant resistance to stress [43,44]. Studying the molecular mechanism of drought resistance and the accumulation mechanism of secondary metabolites in *S. miltiorrhiza* is a prerequisite and a foundation for improving the yield and quality of *S. miltiorrhiza*.

*S. miltiorrhiza* is a model plant among medicinal plants [45]. Its growth is often affected by various types of environmental stress [46,47,48]. Therefore, in-depth research on the regulatory mechanism of stress resistance genes in *S. miltiorrhiza* is of great practical significance. The CCO gene family can regulate plant responses to various types of abiotic stress and has been extensively studied in various plants [49,50,51]; the identification and characterization of CCO based on the whole-genome sequence of *S. miltiorrhiza* were reported for the first time. In this study, 26 CCO members were characterized in *S. miltiorrhiza*. Compared with *A. thaliana* [19], *Glycine max* [52], and *Malus domestica* [49], *S. miltiorrhiza* has a larger CCO family. CCO is a type of double-cleavage diooxygenase that contains the RPE65 domain and generates secondary metabolites during carotenoid metabolism [53]. The secondary metabolites are crucial for carotenoid metabolism and aroma volatiles. Fe^2+^ can activate the catalytic activity of the enzyme [54,55]. The four conserved histidine residues in CCO can regulate their binding to Fe^2+^. In addition, several studies have confirmed that the conservation of CCO protein sequences is not high. Carotenoids are precursors of important plant hormones, such as ABA [56]. The latter plays an important role in many aspects of the plant life cycle, especially in the response of plants to various types of environmental stress [57].

Most studies on the function of the *CCD4* gene focus on plant coloration [58,59,60,61]. However, there is almost no information on the regulation of medicinal active ingredients and drought resistance by *CCD4* in medicinal plants. Therefore, this study first cloned the coding sequence of *SmCCD4* in *S. miltiorrhiza* and found through RT-qPCR that *SmCCD4* responds to ABA induction and PEG stress. Secondly, an overexpression vector for *SmCCD4* was constructed, and a total of eight transgenic strains were obtained. By measuring the content of salvianolic acids and tanshinones in different strains, the *SmCCD4* gene in *S. miltiorrhiza* can upregulate key genes in the biosynthesis pathways of these two secondary metabolites, thereby promoting the accumulation of the compounds. PEG was used to simulate drought stress conditions, and it was found that the ABA content in *SmCCD4* OE strains was significantly increased, and compared with WT strains, the OE strains showed better drought resistance.

## 4. Materials and Methods

### 4.1. Identification of SmCCO Family and Sequence Analysis

The nucleotide and amino acids sequences of the reported *A. thaliana* CCO amino acid sequences *AtCCD1* (AT3G63520.1), *AtCCD4* (AT4G19170.1), *AtCCD7* (AT2G44990.3), *AtNCED2* (AT4G18350.1), *AtNCED3* (AT3G14440.1), and *AtNCED5* (AT1G30100.1) were employed as a query to search against the *S. miltiorrhiza* genome assembly of our lab by the BLASTn algorithm. PFAM (pfam-legacy.xfam.org) analyzed conserved domains, deleted sequences with incomplete RPE65 domains [62], and identified all members of the SmCCO family. We used online websites to once again determine that each member has a complete conserved domain (http://www.ebi.ac.uk/interpro/search accessed on 9 September 2024) and calculate its molecular weight and isoelectric point (https://web.expasy.org/compute_pi/ accessed on 9 September 2024). We used MEGA 6.0 to construct an evolutionary tree for the *S. miltiorrhiza* CCO family by using the neighbor-joining method [63]. We used online websites for motif analysis (http://meme-suite.org/tools/meme accessed on 9 September 2024), gene structure analysis (http://gsds.cbi.pku.edu.cn accessed on 9 September 2024), cis-acting component analysis (http://bioinformatics.psb.ugent.be/webtools/plantcare/ accessed on 9 September 2024), and signal peptide prediction, transmembrane domains, and hydrophilicity/hydrophobicity analysis (http://www.detaibio.com/tools/ accessed on 9 September 2024). We predicted the tertiary structure of the SmCCD4 protein by using the SWISS-MODEL online website (https://swissmodel.expasy.org/ accessed on 9 September 2024).

### 4.2. Plant Materials and Treatments

The cultivation of sterile *S. miltiorrhiza* seedlings was carried out according to the previously described procedure [64]. We treated two-month-old *S. miltiorrhiza* plants by spraying 100 μM ABA solution and 20% PEG8000 solution, while simultaneously spraying the corresponding solvent 10% ethanol solution and water as controls. We collected samples at 0 h, 0.25 h, 0.5 h, 1 h, 3 h, 6 h, 12 h, and 24 h and froze them in liquid nitrogen. Each sample had three biological replicates.

### 4.3. Expression Pattern Analysis

Total RNA isolation kits (Vazyme, Nanjing, China) were used to extract total RNA from the abovementioned plants. The integrity of RNA was detected by agarose gel electrophoresis, and the concentration of RNA was detected by a spectrophotometer. ATGScript Reverse Transcriptase was used to reverse-transcribe total RNA into cDNA (Vazyme, Nanjing, China). The expression pattern of *SmCCD4* was examined by using RT-qPCR and calculated by using the comparative CT method (2^−ΔΔCT^) [65]. By using *SmACT* (DQ243702) [66] as an internal reference, the gene expression levels of different samples were detected. The expression analysis of *SmCCD4* in all samples was performed by using three biological replicates and three technical replicates. Asterisks indicate significance differences from 0 h by using the *t*-test. Data are shown as means ± SDs, n = 3.

### 4.4. SmCCD4 Gene Cloning and Vector Construction

The full-length CDS of *SmCCD4* obtained from the *S. miltiorrhiza* genome database is 1101 bp, and the primers were designed by using Vector NT1Suits (Appendix A). The cDNA of *S. miltiorrhiza* flower was used as the template for PCR amplification. The PCR product was subjected to agarose gel electrophoresis, and the target band of about 1100 bp was cut and recovered by using the agarose gel DNA recovery kit (Vazyme, Nanjing, China). After recycling, the product was measured for concentration by using a spectrophotometer and sequenced for comparison. We constructed overexpression vectors by using Gateway technology [67]. Firstly, a BP reaction was performed to connect the CDS of *SmCCD4* fragments with attB sites to pENTR207 vectors with attP sites, resulting in the production of the pENTR207-SmCCD4 recombinant plasmid. Next, an LR reaction was performed to recombine the pENTR207-*SmCCD4* from the previous step with the target pEarlyGate202 vector containing the attR site, resulting in the overexpression vector pEarleyGate202-*SmCCD4* recombinant plasmid.

### 4.5. Genetic Transformation and Transformant Selection

We transformed the constructed recombinant plasmid pEarleyGate202-*SmCCD4* into *Agrobacterium tumefaciens* EHA105 [64]. We cut the leaves of two-month-old sterile seedlings of *S. miltiorrhiza* into 1 cm × 1 cm squares, with the leaves facing downwards, and placed them on solid MS medium supplemented with 1 µg/mL 6-Benzylaminopurine and 0.1 µg/mL naphthylacetic acid for two days at 25 °C with a 16 h photoperiod at 108 µmol/m^2^/s. The pre-cultured explants were immersed in the diluted bacterial solution (OD 0.2–0.3) for transfection and shook at 25 °C and 100 rpm for cultivation. After 15 min, the explants were taken out to absorb the remaining surface solution and placed on solid 6N medium (25 °C in the dark). Two days later, the explants were put on solid 6N medium (supplemented with 0.1 µg/mL thidiazuron and 50 µg/mL cefotaxime) for callus induction (25 °C under a 16 h photoperiod). We changed the medium every 10 days until callus tissue formed and buds differentiated.

Finally, the buds were transferred from the callus to 1/2 MS solid medium supplemented with 0.1 µg/mL thidiazuron and 50 µg/mL cefotaxime to induce rooting and form complete transgenic plants of *S. miltiorrhiza*. We extracted genomic DNA from different *S. miltiorrhiza* lines and detected positive transgenic lines by amplification using CaMV35S promoter specific primers [68]. Total RNA was also isolated to evaluate gene expression levels through RT-qPCR.

### 4.6. Determination of Tanshinone and Phenolic Acid Contents

The roots of 2-month-old OE-*SmCCD4* and WT *S. miltiorrhiza* plants were harvested and dried at 30 °C until constant weigh. The secondary metabolites were extracted as previously described. An Agilent 1260 HPLC system coupled to an Agilent 6460 QQQ LC/MS system (Agilent, Clara, CA, USA) was used to determine phenolic acid and tanshinone contents under previously described detection conditions [69]. Chromatography was performed on a Welch Ultimate XB-C18 column with a 0.4 mL/min flow rate and a 5 µL injection volume. The mobile phase included Solvent A (acetonitrile) and Solvent B (0.1% formic acid in deionized water), and tanshinones and phenolic acids followed the following gradient distributions: Tanshinones: 0–5 min, A 75–90%, B 25–10%; 5–6 min, A 90–25%, B 10–75%; 6–10 min, A 25%, B 75%. Phenolic acids: 0–6 min, A 20–60%, B 80–40%; 6–7 min, A 60–20%, B 40–80%; 7–10 min, A 20%, B 80%. Analytes were detected in positive ionization mode, and the drying gas flow was 1 L/min.

### 4.7. Physiological Assays

Two-month-old WT, EV, and OE *S. miltiorrhiza* seedlings were analyzed for drought resistance. Superoxide dismutase (SOD), malondialdehyde (MDA), catalase (CAT), and hydrogen peroxide (H_2_O_2_) concentrations were measured by using the Plant Micro Assay Kit (Solarbio, Beijing, China), with three biological and three technical replicates.

### 4.8. Statistics

All experiments were repeated by using three biological replicates and three technical replicates unless otherwise specified. The data were calculated as the means ± standard error of the mean, and significance analysis was performed by using the *t*-test calculated by GraphPad Prism 7.0.4.

## Figures and Tables

**Figure 1 ijms-25-13223-f001:**
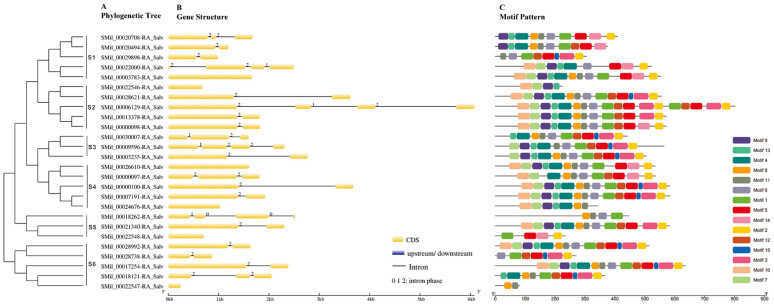
Phylogenetic trees, gene structures, and conserved motifs of SmCCO genes. (**A**) SmCCO proteins were clustered into 6 subgroups, designated as S1 to S6. (**B**) Yellow boxes and grey lines represent exons and introns. (**C**) Boxes represent motif distributions.

**Figure 2 ijms-25-13223-f002:**
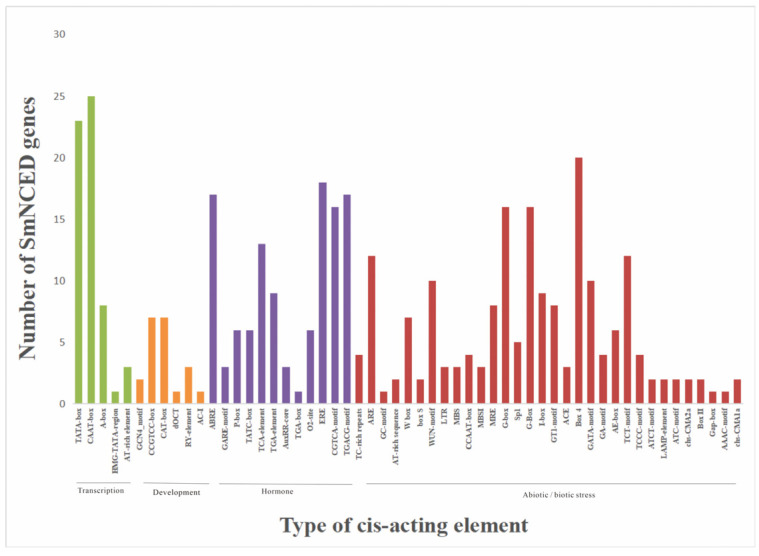
The number of cis-acting elements in the SmCCO promoter. The graph was generated based on the presence of cis-acting elements in response to specific elicitors/conditions/processes (x-axis) in the SmCCO gene family members (y-axis).

**Figure 3 ijms-25-13223-f003:**
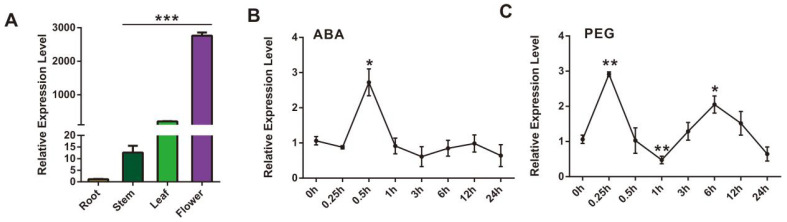
RT-qPCR analysis of *SmCCD4* gene expression level. (**A**) Expression level of *SmCCD4* in different organs (X-axis: root, stem, leaf, and flower) of *S. miltiorrhiza*. (**B**,**C**) Expression level of *SmCCD4* under ABA induction and PEG drought stress. Data were normalized to *SmACT* gene. All data represent means ± SDs of three independent experiments. Statistical significance was assessed with Student’s *t*-test (*** *p* < 0.001; ** *p* < 0.01; * *p* < 0.05). The X-axis represents the treatment time of ABA and PEG.

**Figure 4 ijms-25-13223-f004:**
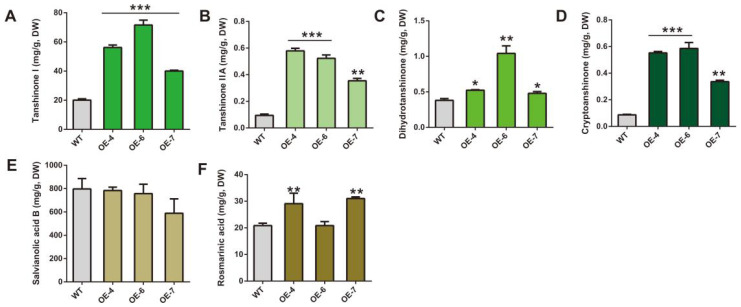
Detection of tanshinone and phenolic acid contents in WT and overexpression strains. (**A**–**D**) Concentration of tanshinone compounds: tanshinone I (**A**), tanshinone IIA (**B**), ihydrotanshinone (**C**), and cryptotanshinone (**D**). (**E**,**F**) Concentration of phenolic acid compounds: salvianolic acid B (**E**) and rosmarinic acid (**F**). All data represent means ± SDs of three independent experiments. Statistical significance was assessed with Student’s *t*-test (*** *p* < 0.001; ** *p* < 0.01; * *p* < 0.05).

**Figure 5 ijms-25-13223-f005:**
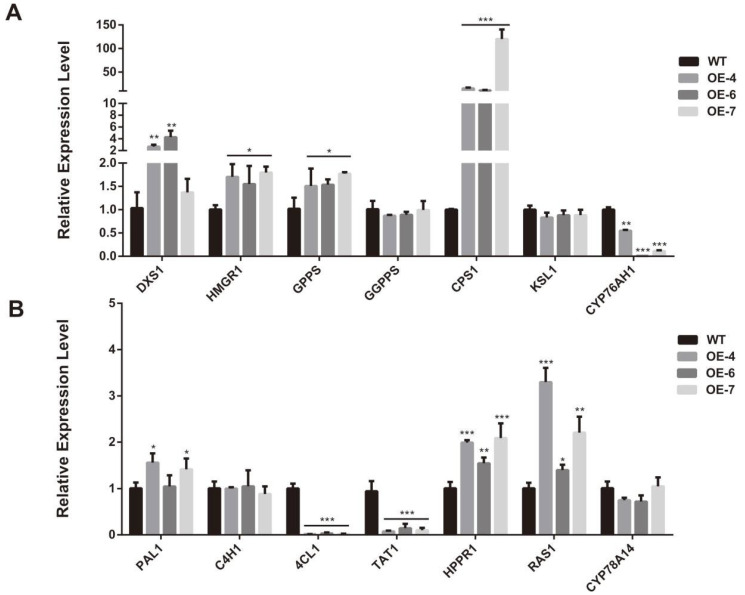
RT-qPCR analysis of key enzyme genes involved in tanshinone (**A**) and phenolic acid (**B**) biosynthetic pathway in overexpression and WT strains. Data were normalized to *SmACT* gene. All data represent means ± SDs of three independent experiments. Statistical significance was assessed with Student’s *t*-test (*** *p* < 0.001; ** *p* < 0.01; * *p* < 0.05).

**Figure 6 ijms-25-13223-f006:**
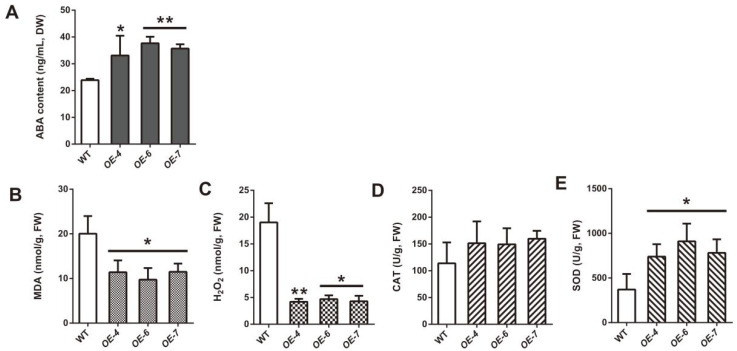
Detection of contents of abscisic acid (**A**) and MDA (**B**), H_2_O_2_ accumulation (**C**), and CAT (**D**) and SOD activity (**E**) in WT and overexpression (OE) strains. All data represent means ± SDs of three independent experiments. Statistical significance was assessed with Student’s *t*-test (** *p* < 0.01; * *p* < 0.05).

**Table 1 ijms-25-13223-t001:** The main medicinal active ingredients in *S. miltiorrhiza*.

Liposoluble Components (Tanshinones)	Water-Soluble Components (Phenolic Acids)
Name	Molecular Structure	Name	Molecular Structure
Tanshinone I	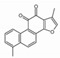	Rosmarinic acid	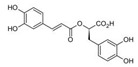
Tanshinone IIA	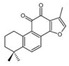	Salvianolic acid A	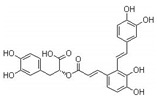
Tanshinone IIB	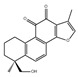	Salvianolic acid B	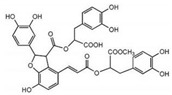
Cryptotanshinone	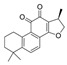	Salvianolic acid C	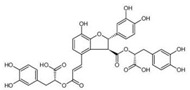
Isotanshinone I	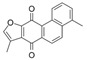	Protocatechuic acid	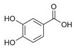

## Data Availability

Data contained within the article.

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
