# Peer review of "Carotenoid Cleavage Dioxygenase Gene CCD4 Enhances Tanshinone Accumulation and Drought Resistance in Salvia miltiorrhiza"

_ijms, 2024, doi:10.3390/ijms252313223_

Round 1
Reviewer 1 Report
Comments and Suggestions for Authors
The research is sound and well designed, the paper is well written and brings light into an important metabolic pathway in a medicinal plant, so we have new knowledge on how nature creates useful plants. The experiments have all the required controls and the results are well analyzed and with statistically significant results.
My minor criticism is that this plant is not familiar to most of the readers, including myself, nor the tanshinones. I would include some explicative figure on the metabolic pathway that authors are investigating, as well as, the chemical strucure of the tanshinones.
Author Response
Dear Reviewer,
We feel great thanks for your professional review work on our manuscript. As you are concerned, there are several problems that need to be addressed. According to your nice suggestions, we have made extensive corrections to our previous draft. The detailed corrections are listed below in the blue text. And the changes marked in red in the revised paper.
Best wishes
Dr. Wen Zhou and Prof. Zhezhi Wang
Reviewer 2 Report
Comments and Suggestions for Authors
Journal IJMS (ISSN 1422-0067)
Manuscript IDijms-3300146
Type Article
Title SmCCD4 enhance tanshinones accumulation and drought resistance in Salvia miltiorrhiza
Authors Qian Tian , Wei Han , Shuai Zhou , Liu Yang , Donghao Wang , Wen Zhou * , Zhezhi Wang *
Section Molecular Plant Sciences
Dear Editor,
I am submitting a review of Manuscript IDijms-3300146.
I have included comments below.
With best regards
Reviewer
TITLE
1.
Develop an acronym in the subject line of the manuscript; not every recipient will be a biologist or academic.
2.
Throughout the manuscript, abbreviations used for the first time should be explained
ABSTRACT
3.
Note the sentence style ’
“The dry root of Salvia miltiorrhiza Bunge which is a perennial herbaceous plant of Lamiaceae 13 Salvia, is one of the staple Chinese herbal medicine in China with long officinal history.”
4.
Please check step by step in the abstract that all structural elements of the abstract have been followed:
(i) put the introduction discussed in a broad context
(ii) highlight the purpose of the study
(iii) briefly describe the main methods used
(iv) summarise the main relevant results
(v) indicate the main conclusions.
KEY WORDS
5.
Please eliminate words occurring in the title of the manuscript. This will increase the searchability of the paper and consequently citation.
INTRODUCTION
l. 31-42.
6.
l. 31 - 33. You cannot lump everything into “one bag”. Instead of mental shortcuts, please fill in the factual data: biological model of the experiment, biologically active chemical compound, dose used, method of administration, time of administration, length of the experiment, main conclusion, etc. The title of the cited original recent scientific publication should refer to the disease entity described. Please amend this.
7.
l. 33 - 35. Instead of keywords, please elaborate on the issues \:
- “significant medicinal value”
- “high conversion efficiency”
- “of secondary metabolites”
broad concepts, please supplement with factual data from the latest scientific literature in exchange for short slogans.
8.
l. 35-42. Please, according to the accepted classification of biologically active compounds, complete the relevant groups of biologically active compounds and the predominant chemical compounds in these groups. I propose to present in the form of a table, quoting the latest scientific publications.
l. 43-56
9.
Please state the role with close reference to the topic of the publication.
“Carotenoid cleavage oxygenase (CCO) can specifically catalyse the cleavage of con-43 jugated double bonds in carotenoid polyene chains, thereby forming various deprotonated carotenoids and their derivatives[6]. According to whether the substrate forms an …”
10.
Please complete what role they play with reference to the subject of your manuscript
“carotenoid cleavage dioxygenase (CCD) “
“9-cis-epoxycarotenoid dioxygenase (NCED)”
11.
l.49-54. Italicise the Latin names of the genus and species. Please use throughout the manuscript and in the literature list.
12.
Please refer to the subject of the manuscript in close connection to the species under study or related species in the family Lamiaceae.
13.
Please complete the year “Wei et al.....”.
l. 57-79.
14.
Please complete the metabolic process “...thereby regulating the colour of plants flowers, fruits,...”
15.
l. 60-62. Please explain the mechanisms of these metabolic processes with reference to the manuscript topic.
16.
l. 63-64. It has been known for years, please explain why with reference to the manuscript topic “...Various abiotic stresses, including drought and high salt, can have adverse effects on plant growth and development...”.
17.
Please complete , which reactions their metabolism with reference to the manuscript topic. “...Research has shown that members of the CCO family are widely involved in various stress responses of plants...”.
18.
l. 66 – 70. Please explain why.
19.
l. 66-79. See comment 11: Brassica oleracea, Arabidopsis thaliana, Lycopersicum esculentum.
20.
Please explain how “...causing changes in physiological and biochemical responses, thereby enhancing the plant's stress resistance...”.
21.
l. 74 - 79. Please complete the metabolic processes indicated, but with close reference to the topic of the manuscript quoting recent literature reports.
l. 80 - 90
22.
“Our research group discovered” Eliminate terms that carry no substantive content. This is a research paper not a review paper.
23.
Please reinforce the rationale for the manuscript topic concept.
24.
Please formulate the specific purpose of the research
RESULTS
25.
Please insert Table 1S (missing from the text).
26.
l. 116 - 120. please check the notation [].
FIGURE 3.
27.
Please complete the description of the X axis
28.
Please insert the letter designation “C” of the third graph.
FIGURE 4, 6
29.
Please complete each diagram with a letter designation.
FIGURE 5.
30.
Please note the Y-axis, should there be a figure 2.0 twice. Please trace this graph again.
DISCUSSION
31.
See comment 11. (l. 229).
32.
l. 229-230. please complete the full species names in Latin in place of the genus names in Latin and English “...Compared to Arabidopsis (9)[9], soybean (8)[24], and apple (21)[25], S. miltiorrhiza has a larger CCO family....”.
33.
Please note and make correction for the double letter notation ‘to Arabidopsis (9)[9], soybean (8)[24], and apple (21)[25]...’
DISSCUSION
34.
l. 213-250.
Please strengthen the discussion by completing the factual information in close reference to the topic of the manuscript. Please increase the number of recent scientific publications cited. Considering the discussion nature of the text instead of the review nature.
35.
Cited all 13 items of literature in the discussion subsection, expect an increase in citation as a result of the latest scientific information on each methodological step .
36.
l. 241-243, not in this section.
37.
Please complete the novelty of the research presented.
38.
Please indicate the applicability of the research results obtained.
39.
Please complete the perspective of the research in the future
MATERIALS AND METHODS
40.
Please complete for each methodological step a literature citation supporting the validity of the research undertaken.
CONCLUSIONS
41.
Please complete the specific conclusions drawn from the research that will address the research objective and the research thesis set.
REFERENCES
42.
Please check each literature item and make corrections according to the authors' guidelines, e.g. item: 7; 8; 12; 15; 18; 26; 34; 37; it is impossible to list them all.

Author Response
Dear Reviewer,
We feel great thanks for your professional review work on our manuscript. As you are concerned, there are several problems that need to be addressed. According to your nice suggestions, we have made extensive corrections to our previous draft. The detailed response are listed below. And the changes marked in red in the revised paper.
Best wishes
Dr. Wen Zhou and Prof. Zhezhi Wang
TITLE
1.Develop an acronym in the subject line of the manuscript; not every recipient will be a biologist or academic.
2.Throughout the manuscript, abbreviations used for the first time should be explained
Response: Please refer to Line 2-12.
ABSTRACT
3.Note the sentence style ’
“The dry root of Salvia miltiorrhiza Bunge which is a perennial herbaceous plant of Lamiaceae 13 Salvia, is one of the staple Chinese herbal medicine in China with long officinal history.”
4.Please check step by step in the abstract that all structural elements of the abstract have been followed:
(i) put the introduction discussed in a broad context
(ii) highlight the purpose of the study
(iii) briefly describe the main methods used
(iv) summarise the main relevant results
(v) indicate the main conclusions.
Response: Please refer to Line 13-26.
KEY WORDS
5.Please eliminate words occurring in the title of the manuscript. This will increase the searchability of the paper and consequently citation.
Response: Please refer to Line 28.
INTRODUCTION
6.l. 31 - 33. You cannot lump everything into “one bag”. Instead of mental shortcuts, please fill in the factual data: biological model of the experiment, biologically active chemical compound, dose used, method of administration, time of administration, length of the experiment, main conclusion, etc. The title of the cited original recent scientific publication should refer to the disease entity described. Please amend this.
7.l. 33 - 35. Instead of keywords, please elaborate on the issues \:
- “significant medicinal value”
- “high conversion efficiency”
- “of secondary metabolites”
broad concepts, please supplement with factual data from the latest scientific literature in exchange for short slogans.
Response: Please refer to Line 33-37.
8.l. 35-42. Please, according to the accepted classification of biologically active compounds, complete the relevant groups of biologically active compounds and the predominant chemical compounds in these groups. I propose to present in the form of a table, quoting the latest scientific publications.
Response: After careful consideration, we believe that it is not necessary to classify and list the secondary metabolites of Salvia miltiorrhiza. Because these contents are already recognized and very familiar to readers.
9.l. 43-56. Please state the role with close reference to the topic of the publication.
“Carotenoid cleavage oxygenase (CCO) can specifically catalyse the cleavage of con-43 jugated double bonds in carotenoid polyene chains, thereby forming various deprotonated carotenoids and their derivatives[6]. According to whether the substrate forms an …”
Response: Please refer to Line 49.
10.Please complete what role they play with reference to the subject of your manuscript
“carotenoid cleavage dioxygenase (CCD) “
“9-cis-epoxycarotenoid dioxygenase (NCED)”
Response: Please refer to Line 53-55.
11.l.49-54. Italicise the Latin names of the genus and species. Please use throughout the manuscript and in the literature list.
Response: Thank you for your feedback. We have checked and revised the entire manuscript. Some of these subfamily names are not specific gene names and do not need to be italicized.
- Please refer to the subject of the manuscript in close connection to the species under study or related species in the family Lamiaceae.
Response: Thank you for your comments. To our knowledge, as well as through database searches, there have been no studies on CCO gene family related genes in other species of the Lamiaceae family. Only Arabidopsis and economic crops such assoybean and grapes have been studied, which we have cited in the introduction.
13.Please complete the year “Wei et al.....”.
Response: Please refer to Line 62.
14.l. 57-79.Please complete the metabolic process “...thereby regulating the colour of plants flowers, fruits,...”
Response: Please refer to Line 64-66.
15.l. 60-62. Please explain the mechanisms of these metabolic processes with reference to the manuscript topic.
Response: Please refer to Line 72-74.
16.l. 63-64. It has been known for years, please explain why with reference to the manuscript topic “...Various abiotic stresses, including drought and high salt, can have adverse effects on plant growth and development...”.
Response: This sentence has caused ambiguity here. After careful consideration, we have removed this sentence.
17.Please complete , which reactions their metabolism with reference to the manuscript topic. “...Research has shown that members of the CCO family are widely involved in various stress responses of plants...”.
Response: As SmCCD4 in the homologous genes BrCCD4 and BoCCD4 in Brassica rapa and Brassica oleracea in the following text (Line 76-80), have metabolic reactions related to the manuscript topic.
18.l. 66 – 70. Please explain why.
Response: The author team unanimously believes that the two references cited in Line 66-70 can already explain the mechanism involved.
19.l. 66-79. See comment 11: Brassica oleracea, Arabidopsis thaliana, Lycopersicum esculentum.
Response: Thank you for your comments. We have checked and revised the entire manuscript for the format, spelling, and grammar errors.
20.Please explain how “...causing changes in physiological and biochemical responses, thereby enhancing the plant's stress resistance...”.
Response: Thank you for your comments. You can refer to the literature cited in this sentence.
21.l. 74 - 79. Please complete the metabolic processes indicated, but with close reference to the topic of the manuscript quoting recent literature reports.
Response: Please refer to Line 87.
- 80 - 90
- “Our research group discovered” Eliminate terms that carry no substantive content. This is a research paper not a review paper.
Response: Thank you for your comments. Please refer to Line 91.
- Please reinforce the rationale for the manuscript topic concept..
- Please formulate the specific purpose of the research
Response: Thank you for your comments. Please refer to Line 91-101.
RESULTS
- Please insert Table 1S (missing from the text).
Response: Please refer to Part 2.1, Line 108, Table S1.
- 116 - 120. please check the notation [].
Response: Please refer to Line 130-131.
FIGURE 3.
- Please complete the description of the X axis
Response: Please refer to Line 160-165.
- Please insert the letter designation “C” of the third graph.
Response: Thank you for your comments. Please refer to Figure 3.
FIGURE 4, 6
- Please complete each diagram with a letter designation.
Response: Thank you for your comments. Please refer to Figure 4 and Figure 6, Line 200-205 and Line 225-228.
FIGURE 5.
- Please note the Y-axis, should there be a figure 2.0 twice. Please trace this graph again.
Response: Thank you for your comments. Please refer to Figure 5.
DISCUSSION
- See comment 11. (l. 229).
Response: Thank you for your comments. We have checked and revised the entire manuscript for the format, spelling, and grammar errors.
- 229-230. please complete the full species names in Latin in place of the genus names in Latin and English “...Compared to Arabidopsis(9)[9], soybean (8)[24], and apple (21)[25], S. miltiorrhiza has a larger CCO family....”.
- Please note and make correction for the double letter notation ‘to Arabidopsis (9)[9], soybean (8)[24], and apple (21)[25]...’
Response: Thank you for your comments. Please refer to Line 245-246.
DISSCUSION
- 213-250.Please strengthen the discussion by completing the factual information in close reference to the topic of the manuscript. Please increase the number of recent scientific publications cited. Considering the discussion nature of the text instead of the review nature.
- Cited all 13 items of literature in the discussion subsection, expect an increase in citation as a result of the latest scientific information on each methodological step .
- 241-243, not in this section.
- Please complete the novelty of the research presented.
- Please indicate the applicability of the research results obtained.
- Please complete the perspective of the research in the future
Response: Thank you for your comments. Please refer to Disscusion Part, Line 230-270.
MATERIALS AND METHODS
- Please complete for each methodological step a literature citation supporting the validity of the research undertaken.
Response: Thank you for your comments. As suggested by the reviewer, we have added more references to support this idea. Please refer to Materials and Methods Part for checking.
REFERENCES
- Please check each literature item and make corrections according to the authors' guidelines, e.g. item: 7; 8; 12; 15; 18; 26; 34; 37; it is impossible to list them all.
Response: I'm very sorry, we didn't understand this comments.
Reviewer 3 Report
Comments and Suggestions for Authors
Comments
Comments and Suggestions for Authors
Dear Author,
It is my pleasure to review the manuscript entitled “SmCCD4 enhance tanshinones accumulation and drought resistance in Salvia miltiorrhiza” a research article submitted to MDPI Journal, IJMS. Authors of this manuscript performed transcriptomic analysis of this plant genome and identified a carotenoid cleavage dioxygenase, SmCCD4, gene is highly responsible for drought stress in S. miltiorrhiza. They identified 26 SmCCO genes in this plant. They checked expression of this gene at various organs of S. miltiorrhiza. Further, they construct SmCCD4 overexpressed transgenic and found it promotes abscisic acid synthesis and also regulate secondary metabolites tanshinones synthesis. The overall experiments, they performed, are well and the results are very convincing and important for cultivation. Thus, the presented results take up an important topic consistent with the profile of the Journal.
I have some suggestions, which might improve the manuscript to make important to the wider readers.
-English should be improved significantly
-Most comments I have made in the original pdf. Please check carefully
-Genes should be italic, throughout the text.
- With more constructive rationale of the study, elaborate clearly in introductory, why this research is necessary and identify gaps of the previously published research.
Results
-in fig.3; expression level indicating peculiar results. Not much changed at all time point, except only just after .5h of treatment. What do you predict from this? You need to confirm is it experimental error or real?
Discussion; This part is not constrictive. You did not discuss your obtained results with comparison to others. Prediction and judgement should be given with proper references to justify results.
Materials and Methods
-Line 263-267; incomplete sentence
-Predicting the tertiary structure of SmCCD4 protein by using the SWISS-MODEL online website (https://swissmodel.expasy.org/).-- ------correct this sentence
-Writing style is not correct. Who have collected the sample?
-Asterisks indicate significance differences from 0 h by using t-test. Data are shown as means ±SD, n=3.---------- This part should be in figure legend

The English could be improved to more clearly express the research.
Author Response
Dear Reviewer,
We feel great thanks for your professional review work on our manuscript. As you are concerned, there are several problems that need to be addressed. According to your nice suggestions, we have made extensive corrections to our previous draft. The detailed corrections are listed below. And the changes marked in red in the revised paper.
Best wishes
Dr. Wen Zhou and Prof. Zhezhi Wang
-English should be improved significantly
Thanks for your suggestion. We invite a colleague who is a native English speaker from the USA to help polish our MS. And we hope the revised manuscript could be acceptable for you. These changes will not influence the content and frame work of the paper. And here we did not list the changes bu tmarked in red in the revised paper. We appreciate for Editors/Reviewers’ warm work earnestly.
-Genes should be italic, throughout the text.
Thank you for your comments. We have checked and revised the entire manuscript for the format, spelling, and grammar errors.
- With more constructive rationale of the study, elaborate clearly in introductory, why this research is necessary and identify gaps of the previously published research.
Thank you for your comments. Please refer to Introduction Part for checking.
Results
-in fig.3; expression level indicating peculiar results. Not much changed at all time point, except only just after .5h of treatment. What do you predict from this? You need to confirm is it experimental error or real?
Due to the fact that we set at least 3 biological replicates for each experiment, there is sufficient data volume to avoid experimental errors. The significant changes after 0.5h of treatment are sufficient to demonstrate the significant response of SmCCD4 to ABA induction and PEG simulated drought stress.
Discussion
This part is not constrictive. You did not discuss your obtained results with comparison to others. Prediction and judgement should be given with proper references to justify results.
Thank you for your comments. Please refer to Discussion Part for checking.
Materials and Methods
-Line 263-267; incomplete sentence
Sorry for the mistake. Please refer to Line 283 for checking. We have checked and revised the entire manuscript for the format, spelling, and grammar errors.
-Predicting the tertiary structure of SmCCD4 protein by using the SWISS-MODEL online website (https://swissmodel.expasy.org/).-- ------correct this sentence
Sorry for the mistake. Please refer to Line 287 for checking. We have checked and revised the entire manuscript for the format, spelling, and grammar errors.
-Writing style is not correct. Who have collected the sample?
Thank you for your comments. Please refer to Line 215-217.
-Asterisks indicate significance differences from 0 h by using t-test. Data are shown as means ±SD, n=3.---------- This part should be in figure legen
Thank you for your comments. Please refer to Figure 4-6 legend.
Round 2
Reviewer 2 Report
Comments and Suggestions for Authors
Journal IJMS (ISSN 1422-0067)
Manuscript ID ijms-3300146
Type Article Title SmCCD4 enhance tanshinones accumulation and drought resistance in Salvia miltiorrhiza
Authors Qian Tian , Wei Han , Shuai Zhou , Liu Yang , Donghao Wang , Wen Zhou * , Zhezhi Wang *
Section Molecular Plant Sciences
Special Issue Molecular Research in Plant Adaptation to Abiotic Stress
Dear Editor
The Authors should respond step by step to each of the Reviewer’s comments.
However, this requirement has not been met, as there are no substantive corrections in the manuscript.
In response to each Reviewer’s comment, the Authors should also directly present fragments that they have added to the manuscript during the revision.
I am sending my first-round review with five suggestions that have not been addressed by the Authors (in blue font with yellow highlighting). I have discontinued checking the text any further.
Here is my explanation to comment 41.
Please check the editing of the bibliography line by line and revise in accordance with the guidelines for authors.
Therefore, in my opinion, the manuscript text cannot be further processed in further editorial stages.
Best regards
Reviewer
Review of manuscript from round one, see comments below (colour blue).
“TITLE
1.Develop an acronym in the subject line of the manuscript; not every recipient will be a biologist or academic.
2.Throughout the manuscript, abbreviations used for the first time should be explained
Response: Please refer to Line 2-12.
ABSTRACT
3.Note the sentence style ’
“The dry root of Salvia miltiorrhiza Bunge which is a perennial herbaceous plant of Lamiaceae 13 Salvia, is one of the staple Chinese herbal medicine in China with long officinal history.”
4.Please check step by step in the abstract that all structural elements of the abstract have been followed:
(i) put the introduction discussed in a broad context
(ii) highlight the purpose of the study
(iii) briefly describe the main methods used
(iv) summarise the main relevant results
(v) indicate the main conclusions.
………………….
Reviewer
The Authors have introduced minimal changes, but forgot about the conclusions.
…………………………………
Response: Please refer to Line 13-26.
KEY WORDS
5.Please eliminate words occurring in the title of the manuscript. This will increase the searchability of the paper and consequently citation.
Response: Please refer to Line 28.
INTRODUCTION
6.l. 31 - 33. You cannot lump everything into “one bag”. Instead of mental shortcuts, please fill in the factual data: biological model of the experiment, biologically active chemical compound, dose used, method of administration, time of administration, length of the experiment, main conclusion, etc. The title of the cited original recent scientific publication should refer to the disease entity described. Please amend this.
7.l. 33 - 35. Instead of keywords, please elaborate on the issues \:
- “significant medicinal value”
- “high conversion efficiency”
- “of secondary metabolites”
broad concepts, please supplement with factual data from the latest scientific literature in exchange for short slogans.
………………….
Reviewer
No response
……………………………………
Response: Please refer to Line 33-37.
8.l. 35-42. Please, according to the accepted classification of biologically active compounds, complete the relevant groups of biologically active compounds and the predominant chemical compounds in these groups. I propose to present in the form of a table, quoting the latest scientific publications.
Response: After careful consideration, we believe that it is not necessary to classify and list the secondary metabolites of Salvia miltiorrhiza. Because these contents are already recognized and very familiar to readers.
………………….
Reviewer
No response; the Authors have ignored important aspects.
…………………………………………………………………
9.l. 43-56. Please state the role with close reference to the topic of the publication.
“Carotenoid cleavage oxygenase (CCO) can specifically catalyse the cleavage of con-43 jugated double bonds in carotenoid polyene chains, thereby forming various deprotonated carotenoids and their derivatives[6]. According to whether the substrate forms an …”
Response: Please refer to Line 49.
………………….
Reviewer
No response; the Authors indicate line 49 „At present, with the extensive development of wild resources, the growth envi-“
,………………………………………..
10.Please complete what role they play with reference to the subject of your manuscript
“carotenoid cleavage dioxygenase (CCD) “
“9-cis-epoxycarotenoid dioxygenase (NCED)”
Response: Please refer to Line 53-55.
………………….
Reviewer
No response; the Authors indicate lines 53-55
……………………………………………….
53 nated carotenoids and their derivatives[6-8]. According to whether the substrate forms an 54 epoxide structure, CCO can be further divided into carotenoid cleavage dioxygenase 55
11.l.49-54. Italicise the Latin names of the genus and species. Please use throughout the manuscript and in the literature list.
Response: Thank you for your feedback. We have checked and revised the entire manuscript. Some of these subfamily names are not specific gene names and do not need to be italicized.
…………………………
Reviewer
No correction for example lines 412, 471, 481, 484, 492, 497, 514, 544.
Lines 428 species name should be lower case.
I do not check further.
……………………………………………………………………
- Please refer to the subject of the manuscript in close connection to the species under study or related species in the family Lamiaceae.
Response: Thank you for your comments. To our knowledge, as well as through database searches, there have been no studies on CCO gene family related genes in other species of the Lamiaceae family. Only Arabidopsis and economic crops such assoybean and grapes have been studied, which we have cited in the introduction.
13.Please complete the year “Wei et al.....”.
Response: Please refer to Line 62.
14.l. 57-79.Please complete the metabolic process “...thereby regulating the colour of plants flowers, fruits,...”
Response: Please refer to Line 64-66.
15.l. 60-62. Please explain the mechanisms of these metabolic processes with reference to the manuscript topic.
Response: Please refer to Line 72-74.
16.l. 63-64. It has been known for years, please explain why with reference to the manuscript topic “...Various abiotic stresses, including drought and high salt, can have adverse effects on plant growth and development...”.
Response: This sentence has caused ambiguity here. After careful consideration, we have removed this sentence.
17.Please complete , which reactions their metabolism with reference to the manuscript topic. “...Research has shown that members of the CCO family are widely involved in various stress responses of plants...”.
Response: As SmCCD4 in the homologous genes BrCCD4 and BoCCD4 in Brassica rapa and Brassica oleracea in the following text (Line 76-80), have metabolic reactions related to the manuscript topic.
18.l. 66 – 70. Please explain why.
Response: The author team unanimously believes that the two references cited in Line 66-70 can already explain the mechanism involved.
19.l. 66-79. See comment 11: Brassica oleracea, Arabidopsis thaliana, Lycopersicum esculentum.
Response: Thank you for your comments. We have checked and revised the entire manuscript for the format, spelling, and grammar errors.
20.Please explain how “...causing changes in physiological and biochemical responses, thereby enhancing the plant's stress resistance...”.
Response: Thank you for your comments. You can refer to the literature cited in this sentence.
21.l. 74 - 79. Please complete the metabolic processes indicated, but with close reference to the topic of the manuscript quoting recent literature reports.
Response: Please refer to Line 87.
- 80 - 90
- “Our research group discovered” Eliminate terms that carry no substantive content. This is a research paper not a review paper.
Response: Thank you for your comments. Please refer to Line 91.
- Please reinforce the rationale for the manuscript topic concept..
- Please formulate the specific purpose of the research
Response: Thank you for your comments. Please refer to Line 91-101.
RESULTS
- Please insert Table 1S (missing from the text).
Response: Please refer to Part 2.1, Line 108, Table S1.
- 116 - 120. please check the notation [].
Response: Please refer to Line 130-131.
FIGURE 3.
- Please complete the description of the X axis
Response: Please refer to Line 160-165.
- Please insert the letter designation “C” of the third graph.
Response: Thank you for your comments. Please refer to Figure 3.
FIGURE 4, 6
- Please complete each diagram with a letter designation.
Response: Thank you for your comments. Please refer to Figure 4 and Figure 6, Line 200-205 and Line 225-228.
FIGURE 5.
- Please note the Y-axis, should there be a figure 2.0 twice. Please trace this graph again.
Response: Thank you for your comments. Please refer to Figure 5.
DISCUSSION
- See comment 11. (l. 229).
Response: Thank you for your comments. We have checked and revised the entire manuscript for the format, spelling, and grammar errors.
- 229-230. please complete the full species names in Latin in place of the genus names in Latin and English “...Compared to Arabidopsis(9)[9], soybean (8)[24], and apple (21)[25], S. miltiorrhiza has a larger CCO family....”.
- Please note and make correction for the double letter notation ‘to Arabidopsis (9)[9], soybean (8)[24], and apple (21)[25]...’
Response: Thank you for your comments. Please refer to Line 245-246.
DISSCUSION
- 213-250.Please strengthen the discussion by completing the factual information in close reference to the topic of the manuscript. Please increase the number of recent scientific publications cited. Considering the discussion nature of the text instead of the review nature.
- Cited all 13 items of literature in the discussion subsection, expect an increase in citation as a result of the latest scientific information on each methodological step .
- 241-243, not in this section.
- Please complete the novelty of the research presented.
- Please indicate the applicability of the research results obtained.
- Please complete the perspective of the research in the future
Response: Thank you for your comments. Please refer to Disscusion Part, Line 230-270.
MATERIALS AND METHODS
- Please complete for each methodological step a literature citation supporting the validity of the research undertaken.
Response: Thank you for your comments. As suggested by the reviewer, we have added more references to support this idea. Please refer to Materials and Methods Part for checking.
REFERENCES
- Please check each literature item and make corrections according to the authors' guidelines, e.g. item: 7; 8; 12; 15; 18; 26; 34; 37; it is impossible to list them all.
Response: I'm very sorry, we didn't understand this comments.”

Author Response
Dear Reviewer,
We greatly appreciate your professional review of our manuscript. The specific response is as follows, and these changes are marked in red in the revised paper.
Best wishes
Dr. Wen Zhou and Prof. Zhezhi Wang
4.Please check step by step in the abstract that all structural elements of the abstract have been followed:
(i) put the introduction discussed in a broad context
(ii) highlight the purpose of the study
(iii) briefly describe the main methods used
(iv) summarise the main relevant results
(v) indicate the main conclusions.
Reviewer
The Authors have introduced minimal changes, but forgot about the conclusions.
Response: Sorry for missing the conclusions. It has been added in the abstract section. Please refer to Line 24-27
INTRODUCTION
6.l. 31 - 33. You cannot lump everything into “one bag”. Instead of mental shortcuts, please fill in the factual data: biological model of the experiment, biologically active chemical compound, dose used, method of administration, time of administration, length of the experiment, main conclusion, etc. The title of the cited original recent scientific publication should refer to the disease entity described. Please amend this.
7.l. 33 - 35. Instead of keywords, please elaborate on the issues \:
- “significant medicinal value”
- “high conversion efficiency”
- “of secondary metabolites”
broad concepts, please supplement with factual data from the latest scientific literature in exchange for short slogans.
Reviewer
No response
Response: Sorry for missing the factual data of medicinal value in Danshen. Please refer to Line 39-49.
8.l. 35-42. Please, according to the accepted classification of biologically active compounds, complete the relevant groups of biologically active compounds and the predominant chemical compounds in these groups. I propose to present in the form of a table, quoting the latest scientific publications.
Response: After careful consideration, we believe that it is not necessary to classify and list the secondary metabolites of Salvia miltiorrhiza. Because these contents are already recognized and very familiar to readers.
Reviewer
No response; the Authors have ignored important aspects.
Response: We have added a table based on your suggestion and classified the components into two categories: liposoluble and water-soluble. Please refer to Table 1.
9.l. 43-56. Please state the role with close reference to the topic of the publication.
“Carotenoid cleavage oxygenase (CCO) can specifically catalyse the cleavage of con-43 jugated double bonds in carotenoid polyene chains, thereby forming various deprotonated carotenoids and their derivatives[6]. According to whether the substrate forms an …”
Reviewer
No response; the Authors indicate line 49 „At present, with the extensive development of wild resources, the growth envi-“
Response: Sorry for pointing to the wrong number of lines. Reference 10-12 are the close reference to this topic.
10.Please complete what role they play with reference to the subject of your manuscript
“carotenoid cleavage dioxygenase (CCD) “
“9-cis-epoxycarotenoid dioxygenase (NCED)”
Reviewer
No response; the Authors indicate lines 53-55
Response: Please refer to Line 64-68.
11.l.49-54. Italicise the Latin names of the genus and species. Please use throughout the manuscript and in the literature list.
Response: Thank you for your feedback. We have checked and revised the entire manuscript. Some of these subfamily names are not specific gene names and do not need to be italicized.
Reviewer
No correction for example lines 412, 471, 481, 484, 492, 497, 514, 544.
Lines 428 species name should be lower case.
I do not check further.
Response: Sorry for the mistakes. We have made revisions to the entire text.
Reviewer 3 Report
Comments and Suggestions for Authors
The authors have addressed all of the comments and suggestions and improved the article substantially. This version is better than previous. However, further revision is needed for mistake free article
Author Response
Dear Reviewer,
We greatly appreciate your professional review of our manuscript. The changes are marked in red in the revised Manuscript.
Best wishes
Dr. Wen Zhou and Prof. Zhezhi Wang
Round 3
Reviewer 2 Report
Comments and Suggestions for Authors
Journal IJMS (ISSN 1422-0067)
Manuscript ID ijms-3300146
Type Article
Title SmCCD4 enhance tanshinones accumulation and drought resistance in Salvia miltiorrhiza
Authors Qian Tian , Wei Han , Shuai Zhou , Liu Yang , Donghao Wang , Wen Zhou * , Zhezhi Wang *
Section Molecular Plant Sciences
Special Issue Molecular Research in Plant Adaptation to Abiotic Stress
Dear Editor
The manuscript was rejected and the reason was explained.
Since the Editor gave the Authors another chance, I expect that the Authors will provide correct responses to each comment in the first review (ROUND ONE) in the following manner:
(i) they will provide a response to each comment (not combining several responses in one)
(ii) under each response to a comment, they will insert the texts that have been added in appropriate parts of the manuscript after the revision (also providing the lines).
(iii) the changes introduced in the manuscript will be marked in red
There are still serious errors, e.g. in the REFERENCES subsection.
Here are the number of erroneous references: 3, 5, 6-9, 13-16, 18-21, 23, 27, 28, 30, 33, 36, 38, 40, 41, 45, 47, 50-53, 55, 58-64, 66, 68-69.
With best regards
Reviewer

Author Response
Dear Reviewer,
We feel great thanks for your professional review work on our manuscript. You provided professional opinions in the Round 1, and we responded almost one by one, with each modification marked in red in the [Manuscript]. There are also some exceptions, such as combining questions 1 and 2, 3 and 4, as they all involve similar issues, so we will make modifications together in the title or abstract.
In the second round, you still emphasized answering each comment one by one. After in-depth discussion, our team carefully responded to each comment and highlighted it in red in the main text. These other reviewers can also clearly see it. However, in the third round, we have already met your requirements for the three questions, but it seems that we still did not satisfy you. Regarding references, they are actually the ones we cite when writing papers or conducting research.
After two comprehensive and meticulous revisions, our team firmly believes that this paper has fully met the high standards required for scientific research papers. Whether in terms of scientificity, ensuring the rigor of research methods and data, or in terms of significance, the potential value of knowledge in related fields.
Sincerely thank you for reviewing our paper amidst your busy schedule and providing valuable feedback. We are well aware of the complexity and subjectivity of paper review. But please believe that our research has unique value and is worth further exploration.
Best wishes
Dr. Wen Zhou and Prof. Zhezhi Wang